# Ivermectin Treatment Response in Onchocerca Volvulus Infected Persons with Epilepsy: A Three-Country Short Cohort Study

**DOI:** 10.3390/pathogens9080617

**Published:** 2020-07-29

**Authors:** Alfred Dusabimana, Dan Bhwana, Stephen Raimon, Bruno P. Mmbando, An Hotterbeekx, Floribert Tepage, Michel Mandro, Joseph N. Siewe Fodjo, Steven Abrams, Robert Colebunders

**Affiliations:** 1Global Health Institute, University of Antwerp, Doornstraat 331, 2610 Antwerp, Belgium; alfred.dusabimana@uantwerpen.be (A.D.); an.hotterbeekx@uantwerpen.be (A.H.); JosephNelson.SieweFodjo@uantwerpen.be (J.N.S.F.); Steven.Abrams@uantwerpen.be (S.A.); 2National Institute Medical Research, Tanga Centre, P.O. Box 5004 Tanga, Tanzania; danbhwana@yahoo.com (D.B.); b.mmbando@yahoo.com (B.P.M.); 3Amref South Sudan, P.O. 30125 Juba, South Sudan; Stephen.Jada@amref.org; 4Ministry of Health, Bas Uélé province, B.P. 105 Buta, Democratic Republic of Congo; floritepage@yahoo.fr; 5Provincial Health Division Ituri, Ministry of Health, Bunia, P.O. Box 57 Ituri, Democratic Republic of Congo; Michel.MandroNdahura@student.uantwerpen.be; 6Interuniversity Institute for Biostatistics and statistical Bioinformatics, Data Science Institute, Hasselt University, 3590 Diepenbeek, Belgium; 7Robert Colebunders, Global Health Institute, Gouverneur Kinsbergencentrum, University of Antwerp, Doornstraat 331, 2610 Wilrijk, Belgium

**Keywords:** ivermectin, sub-optimal response, onchocerciasis, *Onchocerca volvulus*, epilepsy

## Abstract

Despite a long history of community-directed treatment with ivermectin (CDTI), a high ongoing *Onchocerca volvulus* transmission is observed in certain onchocerciasis-endemic regions in Africa with a high prevalence of epilepsy. We investigated factors associated with higher microfilarial (mf) density after ivermectin treatment. Skin snips were obtained from *O. volvulus-*infected persons with epilepsy before, and 3 to 5 months after ivermectin treatment. Participants were enrolled from 4 study sites: Maridi (South Sudan); Logo and Aketi (Democratic Republic of Congo); and Mahenge (Tanzania). Of the 329 participants, 105 (31.9%) had a post-treatment mf density >20% of the pre-treatment value. The percentage reduction in the geometric mean mf density ranged from 69.0% (5 months after treatment) to 89.4% (3 months after treatment). A higher pre-treatment mf density was associated with increased probability of a positive skin snip after ivermectin treatment (*p* = 0.016). For participants with persistent microfiladermia during follow-up, a higher number of previous CDTI rounds increased the odds of having a post-treatment mf density >20% of the pre-treatment value (*p* = 0.006). In conclusion, the high onchocerciasis transmission in the study sites may be due to initially high infection intensity in some individuals. Whether the decreasing effect of ivermectin with increasing years of CDTI results from sub-optimal response mechanisms warrants further research.

## 1. Introduction

With an estimated 20.9 million people infected worldwide, of whom over 99% live in sub-Saharan Africa [1], *Onchocerca volvulus (O. volvulus)* infections are an important global health problem. *O. volvulus* is a filarial nematode causing onchocerciasis and is transmitted by blackflies (*Simuliidae*). Clinical manifestations associated with onchocerciasis include skin and eye disease, nodding syndrome, and other forms of epilepsy (hereafter referred to as onchocerciasis-associated epilepsy (OAE)) [2]. Adult female worms reside in subcutaneous nodules and produce larvae, called microfilariae (mf), that migrate under the skin.

Ivermectin, a broad-spectrum anti-helmintic drug, rapidly kills mf. A single dose of ivermectin leads to a decline in skin mf density of approximately 99% within the first two months [3]. Unfortunately, ivermectin does not kill the adult worms, it only temporarily represses mf production by the adult female worms and therefore mf production resumes at a slow rate after approximately 3 to 6 months [4,5]. Nonetheless, mf loads are expected to remain below 20% of the pre-ivermectin levels for up to 10 months following a single dose of ivermectin [3]. Therefore, many onchocerciasis endemic areas have adopted annual or bi-annual community-directed treatment with ivermectin (CDTI) as a strategy to decrease onchocerciasis transmission [6,7].

Onchocerciasis elimination programs using CDTI have significantly reduced *O. volvulus* transmission and onchocerciasis-related blindness in many African countries [7]. Nevertheless, despite a long history of CDTI, there is still active onchocerciasis transmission in many endemic areas in Ghana [8], Cameroon [9], Democratic Republic of Congo (DRC) [10], and Tanzania [11]. Moreover, such meso- and hyperendemic settings have also been noted to harbor a high burden of OAE [10,11,12,13]. In Ghana and Cameroon, an *O. volvulus* phenotype with sub-optimal ivermectin response has been documented [4,14] which may contribute, at least partly, to the persistence of *O. volvulus* transmission. To determine host factors associated with the high onchocerciasis prevalence and transmission observed in OAE hotspots, we conducted a study assessing individual mf density before and after ivermectin administration.

## 2. Materials and Methods

### 2.1. Study Participants and Study Sites

We performed the study among persons with epilepsy (PWE) with the intention to investigate the effect of ivermection on microfilarial (mf) density but also to evaluate whether ivermectin could decrease the frequency of seizures in PWE infected with *O. volvulus.* Study participants were PWE that were identified during door-to-door surveys in four onchocerciasis endemic study sites. The four study sites were the Aketi health zone, Bas Uélé province and the Logo health zone, Ituri province in the Democratic Republic of Congo (DRC), Mahenge, Ulanga district in Tanzania and Maridi in South Sudan. In Maridi, Aketi and Mahenge participants were enrolled only for a short follow up study, while in the Logo health zone, the 132 participants were enrolled in a randomized clinical trial to evaluate the potential added value of ivermectin in decreasing the frequency of seizures in PWE with *O. volvulus* infection treated with phenobarbital [15]. Right and left skin snips were obtained from each participant before ivermectin treatment intake. Participants with detectable mf in their skin snips were asked to have the skin snip testing repeated 3 to 5 months after ivermectin treatment. Results of the effect of ivermectin on the frequency of seizures will be published elsewhere. All study sites had a different history of CDTI and a different schedule of skin snip testing, as depicted in Table 1.

### 2.2. Aketi Health Zone, DRC

The Aketi health zone is an onchocerciasis hyper-endemic area in Bas-Uélé province in the DRC where CDTI was initiated 14 years ago [16]. Despite this long history of CDTI, an epilepsy prevalence of 5.7% was reported in Aketi rural town during a door-to-door household survey in 2017. In addition, a seroprevalence of OV16 IgG4 antibodies against *O. volvulus* of 64.5% was reported in children aged 7–10 years old, suggesting a high ongoing onchocerciasis transmission [10]. In January 2018, skin snips were obtained from selected PWE from Wela, Makoko, and Aketi rural town. Eighty-one skin snip positive PWE [17] were asked to participate in a follow up study and to be skin snipped again three months after ivermectin intake. Post-ivermectin skin snips could be obtained from 74 of these PWE.

### 2.3. Logo Health Zone, DRC

In October 2017, a proof-of-concept randomized clinical trial investigating the effect of ivermectin on seizure frequency in *O. volvulus* infected PWE was initiated in the Logo health zone, an onchocerciasis-endemic area with a high epilepsy prevalence (4.6%) where ivermectin had never been distributed before [13]. In February 2018, PWE were asked to participate in a randomized clinical trial to evaluate the effect of ivermectin on the frequency of seizures in *O. volvulus* infected PWE [15]. In 392 PWE, skin snips were obtained and 143 (36.5%) of them had detectable mf [17]. These 143 skin snip positive PWE were asked to participate in a follow up study and to be skin snip tested again four months after ivermectin intake. In 136 (95%) of them, pre-and post-ivermectin skin snip results were obtained.

### 2.4. Maridi, South Sudan

In May 2018, a door-to-door household survey in eight villages in an onchocerciasis endemic area in Maridi county showed an overall epilepsy prevalence of 4.4%, reaching up to 11.9% in the Kazana II village, which is located closest to the Maridi dam, a blackfly breeding site [12]. CDTI had been interrupted for at least ten years in this region and was re-initiated in 2017. During a CDTI round in December 2018, PWE identified in the initial survey were asked to participate in a study to determine the prevalence of *O. volvulus* infection among PWE. Of the 318 PWE who agreed to participate, 270 (84.9%) had detectable mf in their skin snips pre-ivermectin [18]. These 270 skin snip positive PWE were asked to participate in a follow up skin snip study 5 months after ivermectin intake. In 102 (38%) of them, pre- and post-ivermectin skin snip results were obtained.

### 2.5. Mahenge, Tanzania

Located in the Ulanga district, Mahenge is a known onchocerciasis-endemic focus. The Tanzanian National Onchocerciasis Control Programme initiated annual CDTI since 1997 and bi-annual CDTI in 2019. In January 2017, we conducted an epilepsy prevalence survey in two rural villages in the Mahenge area (Mdindo, Msogezi), and documented a high prevalence of epilepsy (3.5%) among the general population in these villages as well as a high prevalence of OV16 antibodies in children 6–10 years (42.5%) [11]. PWE identified during the initial survey were asked to participate in a study to determine the prevalence of *O. volvulus* infection among PWE. Of the 42 PWE who agreed to participate, 22 (52.4%) presented mf in skin snips (Bhwana D., personal communication). In April 2019, these 22 PWE with positive skin snip were asked to participate in a follow up study which entailed collecting a second skin snip 3 months after ivermectin intake. In 17 (77%) of them, pre- and post-ivermectin skin snip results were obtained.

#### 2.5.1. Participant Data

Upon enrollment in the study, we obtained the following data from all study participants: age, gender, weight, height, and town or village of residence. In addition, we obtained from the health authorities the exact number of previous rounds of CDTI in each study site (Table 1).

#### 2.5.2. Skin Snip Testing

Skin snips were obtained from each posterior iliac crest of eligible participants using a Holt-type punch. Snips were immediately placed in two wells of a microtitre plate containing 3 drops of normal saline solution and incubated for 24 hours at room temperature to allow mf to emerge into the fluid. After the incubation period, mf in the solution was examined microscopically under a x40 magnification and counted by a trained technician. Mf densities were expressed as arithmetic means between mf count at right skin snips and mf count at left skin snips. One punch was used per subject and punches were sterilized between subjects using steam under pressure (autoclave).

#### 2.5.3. Ivermectin Treatment

Ivermectin was given under direct observation at approximately 150 µg per kg of body weight or using a height equivalent, as prescribed by the World Health Organization guidelines [6]. Our primary indicator of ivermectin treatment efficacy was the reduction in skin mf densities in *O. volvulus*-infected PWE measured 3–5 months after ivermectin treatment, both at the individual level and per study site. We also investigated host-factors associated with a persistent microfiladermia in the post-ivermectin skin snip samples.

#### 2.5.4. Data Processing

The mf density of participants before and after ivermectin treatment were entered into electronic spreadsheets. Given that mf density was recorded as mf per skin snip, and that mf density per skin snip in a small body surface area (BSA) (mostly for children) is likely to be higher than in a large BSA, we calculated participants’ BSA using Boyd’s formula:$$BSA\;\left(in\;m^{2}\right)=0.0003207*W^{\left(0.7285-0.0188\log\left(W\right)\right)}*H^{0.3},$$
where W = weight of the participant (in kg) and H = height of the participant (in m). We then used the individual BSA to normalize individual *O. volvulus* skin mf densities [19,20]. A BSA of 1.73 is mostly used in medical physiology to normalize a number of physiological variables [21]. The absolute value of individual pre-ivermectin skin snip mf density was therefore normalized as: pre-/post-ivermectin skin mf density*individual rBSA with relative BSA (rBSA) equal to BSA/1.73.

#### 2.5.5. Statistical Analysis

Baseline characteristics of all participants within each study site were summarized using the absolute and relative frequency for categorical variables, and median (with interquartile range, IQR) for continuous variables. Mf density before and after ivermectin for each study site were summarized with geometric mean (GM) calculated as the *n*th root of the product of individual mf density (to which 1 was added so as to include post-ivermectin negative skin snips), with *n* equal to number of PWE per study site. The percentage reduction in GM mf density per study site was calculated as the difference between pre- and post-ivermectin GM mf density divided by pre-ivermectin GM density multiplied by 100. The distribution of post-ivermectin *O. volvulus* mf density was skewed to the right and a large proportion of *O. volvulus* mf density values was zero. Therefore, post-ivermectin *O. volvulus* mf density data were modeled using Poisson and negative binomial hurdle models and zero-inflated Poisson and negative binomial models [22]. Akaike’s Information Criteria (AIC) was used to select the ‘best’ model among all candidate models [23]. Parameter estimation in the different models was done using maximum likelihood estimation. In a hurdle model, the analysis consists of a separate logistic regression model and a truncated Poisson (or negative binomial) regression model. The logistic regression model is used to assess the effect of predictors on the probability of having a positive skin snip after ivermectin treatment, whereas the truncated Poisson (or negative binomial) regression approach studies the effects of covariates on post-ivermectin *O. volvulus* mf densities conditional on having a positive skin snip after ivermectin treatment. All possible two-way interactions between baseline characteristics were tested to be included in the final model using the likelihood ratio test. Data were analyzed using SAS 9.4 (SAS Institute Inc.) and two-sided tests at a significance level of 5%.

## 3. Results

A total of 329 *O. volvulus-*infected PWE from four sites participated in the study (Table 2). Study sites were comparable in terms of gender and age, with the exception of Mahenge where participants appeared to be older (Kruskal–Wallis chi-square test = 41.8, and degree of freedom (df) = 3, *p* < 0.001)

### 3.1. Effects of Ivermectin Treatment on the Mf Density of Participants

Of the 329 PWE, 105 (31.9%) experienced a <80% decrease in mf density in their post-treatment skin snips (implying that the post-treatment mf density was >20% of the pre-treatment value). The proportion of PWE with <80% decrease in mf density increased with the number of months before the second skin snip was obtained (Mantel–Haenszel chi-square: 48.8, df =1, *p* < 0.001), being highest in Maridi (second skin snips after 5 months) and lowest in Aketi (second skin snips after 3 months). At the community level, reduction in geometric mean mf density was highest in the Aketi study site (89% reduction) followed by Logo (86% reduction) (Table 3).

### 3.2. Host Factors Associated with Post-Ivermectin Mf Density Values Given a Post-Ivermectin Positive Skin Snip Test

To investigate host factors associated with post-ivermectin *O. volvulus* mf density, we described the average post-ivermectin mf density value, conditional on having a positive skin snip after ivermectin, while adjusting for gender, age, height, number of previous CDTI rounds in the site, duration since last ivermectin intake, and mf density before ivermectin treatment. The negative binomial hurdle model, for which results are presented in Table 4, outperformed other hurdle and zero-inflated models in terms of AIC (Table A1). A higher pre-ivermectin *O. volvulus* mf density was associated with an increased probability of having detectable mf in the follow-up skin snip (Table 4). The effect of pre-ivermectin *O. volvulus* mf density did not change when BSA was included as an offset in the model.

In Figure 1, we graphically display the quadratic effect of age on the probability of a positive skin snip after ivermectin treatment. More specifically, a plot of age-dependent odds ratio for a positive follow-up skin snip reveals that 38 years is the transition age before which the probability of a positive skin snip after ivermectin treatment decreases with increasing age, whereas the reverse is true for individuals aged 38 years or more (Figure 1).

A higher number of previous CDTI rounds in the study site and a longer time lapse between ivermectin intake and the follow-up skin snip increased the odds for <80% individual skin mf reduction (Table 5).

## 4. Discussions

In this study, we investigated treatment response to ivermectin in 329 *O. volvulus-*infected PWE living in four onchocerciasis-endemic areas in sub-Saharan Africa. The post-ivermectin mf density was >20% the pre-treatment value in 105 (31.9%) participants, suggesting that their response to ivermectin warrants further investigation [8]. We observed the highest GM mf reduction of 89.4% three months after ivermectin treatment, as compared to 69.0% GM mf reduction five months post-treatment. Previous studies reported up to 98% reduction of GM mf density, 14 to 90 days after a single dose of ivermectin, with a recuperation of adult worm fertility starting around the third month resulting in a reduction in GM mf density of about 84%, four months post-ivermectin [3].

In the three study sites where the second skin snip was obtained within four months, the percentage of PWE with positive skin snips after ivermectin was lower when compared with a study in Ghana, where 70% of participants had detectable mf four months post-ivermectin [8]. This was not the case with participants from Maridi (South Sudan) whose skin snips were obtained 5 months after treatment and up to 80.4% of participants still presented with microfiladermia. The proportion of participants with positive skin snips post-treatment was higher in our study than observed in moxidectin-treated participants during a comparative trial with ivermectin in the Logo health zone. In this trial, only 8% of the participants who received moxidectin still had detectable mf six months after treatment [24]. In addition to being more potent in reducing mf density, the microfilaricidal effects of moxidectin also last longer than ivermectin [25]. Therefore annual moxidectin or bi-annual ivermectin treatment should be considered for mass treatment in hyperendemic settings as it results in a long-lasting suppression of mf compared to ivermectin once a year.

In our study, we observed that a high pre-ivermectin mf density was significantly associated with a lower mf reduction in the follow-up skin snip. This was previously reported by Pion et al. [26] and in a meta-analysis by Churcher et al. [27], possibly because of the presence of a higher adult worm burden and/or higher numbers of newly patent adult worms in individuals with higher mf density. Therefore, we assume that almost all of the post-treatment positive skin snips in our study can be explained by skin repopulation, which is in line with our observation that post-ivermectin skin snip positivity increased with increasing time between ivermectin treatment and follow-up skin snipping.

We found that in younger persons, the post-ivermectin *O. volvulus* mf densities initially decreased with an increase in age as previously documented by Pion et al. [26]. A potential explanation is that skin repopulation may be more rapid in younger individuals because the adult worms are still maturing (lifespan of about 9–11 years) and rapidly releasing huge numbers of mf in the early phase of the infection [26,28]. In the older participants (>38 years) however, post-ivermectin *O. volvulus* mf densities increased as they got older (Figure 1). A possible explanation for the reversal in trends observed after 38 years could be the absence of adjustments for the number of nodules in our multivariate model [26,29].

An increasing number of previous CDTI rounds was associated with higher probability of achieving <80% mf reduction after ivermectin treatment (Table 5). This finding concurs with previous reports from Cameroon [26,29] where the embryostatic effect of ivermectin was reduced among individuals who had been treated several times with ivermectin in the past, compared to ivermectin-naïve individuals. In contrast, a longer history of CDTI tended to decrease the chances of having a positive skin snip post-ivermectin, although this trend was not statistically significant (*p* = 0.079; Table 4). This could be due to a very low pre-treatment mf density among participants who had been receiving ivermectin for several years prior to our study.

Our study has several limitations. Firstly, our study sites were very different in many aspects, including the time between ivermectin treatment and follow-up skin snipping, and the number of previous CDTI rounds. Secondly, we only evaluated mf densities at two time points and given that we did not collect adult female worms at either of the four sites, we were not able to assess the fecundity and mf density dynamics immediately after ivermectin use, which are needed to confirm sub-optimal treatment response. Moreover, the number of palpable nodules, as a proxy for the total number of adult worms, was not assessed in our study participants. Finally, PWE who take anti-epileptic drugs may experience decreased ivermectin drug levels. It is well known that phenobarbital can influence the *p*-glycoprotein (MDR1) transporter, which plays an important role in the elimination of ivermectin. Unfortunately, ivermectin plasma concentrations were not measured in this study.

In conclusion, our study shows that ivermectin effectively reduces mf density in our study participants, similar to what was observed in other onchocerciasis endemic areas. The fact that almost one third of our participants still had >20% of their pre-treatment mf density, as well as the high ongoing *O. volvulus* transmission observed at our study sites are most likely related to elevated mf densities at baseline as a result of ineffective or inexistent onchocerciasis elimination programs. While we clearly demonstrate that post-ivermectin parasitic load depends on pre-treatment infection intensity, it is unclear whether some study participants exhibit a sub-optimal response to ivermectin. Resistance to the ivermectin treatment has been reported in several parasites in veterinary studies [30,31,32]. Given the rapid rise and spread of ivermectin resistance in the veterinary field, studies should also investigate the possibility of ivermectin resistance in human medicine. In the light of the apparent decreasing effect of ivermectin as the number years of CDTI increases, studies need to investigate the number/fecundity of adult female worms that may be contributing to skin repopulation, and whether a sub-optimal response genotype is present in these areas as was the case in Ghana [14].

## Figures and Tables

**Figure 1 pathogens-09-00617-f001:**
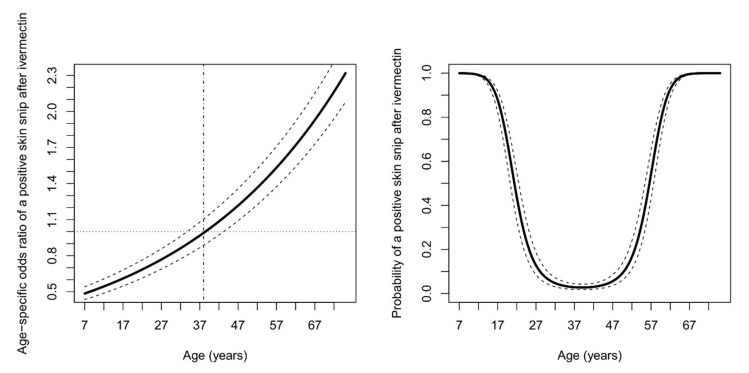
The age-dependent odds ratio of having a positive skin snip after ivermectin for a unit increase in age (left panel). Probability of having a positive skin snip after ivermectin treatment as function of age for female persons of height 180 cm with more than a year since last CDTI round, with 175 mf per skin snip before ivermectin and living in a community where CDTI has been implemented for at least 10 years (right panel). Solid black lines represent the point estimates and dashed lines indicate pointwise 95% confidence bands based on the delta method.

**Table 1 pathogens-09-00617-t001:** Years of community-directed treatment with ivermectin (CDTI) and timing of skin snip testing per study site.

Characteristic	South Sudan	DRC		Tanzania
	Maridi	Logo	Aketi	Mahenge
Years of CDTI	1	0	14	21
Number of months since last CDTI	12	NA	12	5
Number of months between pre-treatment skin snip (followed by ivermectin administration) and post-treatment skin snip	5	4	3	3

NA = not applicable because of no previous CDTI round.

**Table 2 pathogens-09-00617-t002:** Characteristics of study participants in the four study sites.

Participant Characteristic	South Sudan	DRC		Tanzania
	Maridi (*n* = 102)	Logo (*n* = 136)	Aketi (*n* = 74)	Mahenge (*n* = 17)
Age (years), Median (IQR)	16.5 (14.0–18.0)	22.0 (17.0–30.5)	17.0 (15.0–20.0)	35.0 (25.0–45.0)
Male gender: *n* (%)	53 (50)	69 (49)	40 (49)	11 (46)
Weight (kg), Median (IQR)	38.0 (30.0–48.0)	45.0 (38.5–50.0)	37.5 (43.5–54.5)	45.0 (40.0–51.0)
Height (cm), Median (IQR)	158.0 (154.5–162.0)	154.0 (147.0–160.0)	154.0 (138.0–156.0)	154.0 (146.0–160.0)
Pre-ivermectin mf density per skin snip, Median (IQR)	22.7 (12.0–44.5)	28.0 (7–93.0)	9.4 (3.0–52.5)	2 (2.0–20.6)
Pre-ivermectin mf density, GM (SE)	15.0 (1.1)	24.7 (3.2)	12.9 (2.1)	5.7 (1.6)

N: number; mf: microfilariae; IQR = Interquartile range; GM: Geometric mean; SE: Geometric mean standard error.

**Table 3 pathogens-09-00617-t003:** Post-ivermectin parasitological data of study participants.

	Maridi (*n* = 102)	Logo (*n* = 136)	Aketi (*n* = 74)	Mahenge (*n* = 17)
Months post-ivermectin intake: *n*	5	4	3	3
Positive skin snip: *n* (%)	82 (80)	66 (48)	18 (24)	9 (53)
<80% decrease in individual mf density: *n* (%)	52 (51)	38 (28)	8 (11)	7 (41)
Overall Mf per skin snip, Median (IQR)	4.5 (1.0–17.5)	0.0 (0.0–9.0)	0.0 (0.0–0.5)	0.5 (0.0–1.0)
Mf per skin snip of PWE with post-ivermectin positive skin snip; Median (IQR)	5.0 (1.5–19.0)	10.5 (2.5–38.0)	2.0 (0.5–5.5)	1 (0.5–1.0)
Post-ivermectin GM mf density per skin snip, GM (SE)	4.7 (0.6)	3.4 (0.5)	1.4 (0.2)	1.6 (0.3)
Overall percentage reduction in GM mf density per skin snip (%)	69	86	89	72

*n*: number; mf: microfilariae density, IQR = Interquartile range; GM: Geometric mean; SE: Geometric mean standard error.

**Table 4 pathogens-09-00617-t004:** Negative binomial hurdles model to assess host factors associated with post-ivermectin *O. volvulus* microfilarial density.

	Post-Ivermectin *O. volvulus* Mf Density				Probability of Positive Skin Snip after Ivermectin Treatment			
Factors	Coeff	95% CI		*p*-Value	Coeff	95% CI		*p*-Value
Female gender	−0.312	−0.946	0.323	0.336	−2.518	−4.913	−0.124	0.039
Male gender (reference)								
Age (years)	0.012	−0.041	0.064	0.661	−0.391	−0.656	−0.125	0.004
Age*age (years)	−0.001	−0.003	0.001	0.349	0.012	0.003	0.020	0.007
Height in cm	−0.007	−0.055	0.041	0.773	0.091	−0.010	0.192	0.076
Number of previous CDTI rounds in site	−0.216	−0.515	0.083	0.157	−0.591	−1.252	0.069	0.079
>1 year since last ivermectin dose	0.455	−0.476	1.386	0.338	−2.020	−4.137	0.097	0.062
<1 year since last ivermectin dose (reference)								
Follow-up skin snip at month 3	2.734	−1.823	7.290	0.240				
Follow-up skin snip at month 4	0.533	−0.384	1.450	0.255				
Follow-up skin snip at month 5 (reference)								
Pre-ivermectin microfilarial density ^#^	0.337	0.064	0.611	0.016	1.187	0.161	2.213	0.023

#: Log-transformed; Coeff = Negative binomial hurdle model coefficients, CI = Confidence interval; Age*age = Quadratic effect of age.

**Table 5 pathogens-09-00617-t005:** Logistic regression to assess the factors associated with *O. volvulus* mf density reduction of <80% following ivermectin treatment in the four study sites.

Factors	Coeff	95% CI		*p*-Value
Female gender	−0.058	−0.568	0.453	0.825
Male gender (reference)				
Age (years)	−0.097	−0.192	−0.002	0.044
Age*age (years)	0.001	0.001	0.003	0.045
Height in cm	−0.019	−0.048	0.010	0.207
Number of previous CDTI rounds in site	0.282	0.080	0.484	0.006
Follow-up skin snip at month 3	−6.943	−10.064	−3.822	<0.001
Follow-up skin snip at month 4	−1.101	−1.850	−0.351	0.004
Follow-up skin snip at month 5 (reference)				
Pre-ivermectin microfilarial density ^#^	0.202	0.012	0.392	0.037

#: Log-transformed; Coeff = Logistic regression model coefficients, CI = Confidence interval; Age*age = Quadratic effect of age.

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
