# Peer review of "Ivermectin Treatment Response in Onchocerca Volvulus Infected Persons with Epilepsy: A Three-Country Short Cohort Study"

_pathogens, 2020, doi:10.3390/pathogens9080617_

Round 1
Reviewer 1 Report
This manuscript describes a study of effect of ivermectin on microfilarial density of Onchocerca volvulus in infected patients who also have epilepsy.
A higher number of previous CDTIs (community-directed treatment with ivermectin) and past years were found to be associated with a higher proportion of patients with more than 20% microfilarial density (=less than 80% reduction) after treatment. The authors hypothesize that this is due “to elevated base-line microfilarial density levels as a result of ineffective or inexistent onchocerciasis elimination programmes”.
Although onchocerciasis is characterized by a variable clinical spectrum in which epilepsy is an associated feature and not an obligatory criterion, the study was conducted by a primary selection of epilepsy patients.
- What were the reasons for this preselection criterion?
- Perhaps the data from the four study sites have been produced in previous studies and were reused for this specific study? Please specify.
- In which time period (year, month) was the study conducted in each of the four sites?
- This information is needed in order to have insight in how much data from successive publications are overlapping, also in view avoiding multiple inclusions in future meta-analyses.
Table 3 and corresponding text in Results needs clarification or correction.
- The second line in this table 3 specifies a total of 329 persons with epilepsy (PWE) included into this study, divided over the four study sites: 102, 136, 74, 17 respectively.
- The third line specifies how many of these PWEs had a positive skin snip:
- 82 (80,4%), 66 (48,5%), 18 (24,3%), 9 (52,9%) respectively.
- The fourth line specifies the number of these with less than 80% decrease (=more than 20% remaining level) of microfilarial density [after treatment]:
- 52 (50,9%), 38 (27,9%), 8 (10,8%, 7 (41,2%) respectively).
- Apparently, the denominators used for calculation of percentages this fourth line is the total number of PWE (second line) and not those PWE who had a positive skin snip (fourth line). When the correct denominators from the second line are used, the percentages would be as follows:
- 52 (63%), 38 (58%), 8 (44%), 7 (78%) respectively, and without decimal.
- This result would completely change the amount of difference between the four study sites!
- It also raises the question as to how this may have had an effect on all the subsequent complex statistical analyses which by, its nature, are already difficult to comprehend by many readers.
- Furthermore, since the denominators are each below 100, percentages should not be presented with one decimal suggesting high precision of the estimate, but without a decimal.
In the conclusion, the authors propose that the relatively high proportion of persons showing less than 80% reduction of microfilarial density and its association with base-line levels of mf density and number and years of previous CDTI may be explained by higher pretreatment base-line levels. A more plausible explanation is the occurrence and induction of resistance by previous treatment rounds. Occurrence of ivermectin resistance in parasitic disease in animals and humans is increasingly recognized, and this may play a role not only at the level of the parasite i.c. Onchocerca volvulus, but also at its endosymbiont Wolbachia. Deep sequencing of both Onchocerca volvulus and Wolbachia and quantification of genetic variants may be one of many ways to move forward in order to unravel the biological mechanisms of apparent development of ivermectin resistance. Three selected references on this are:
- Sulaiman et al., Indian J Med Res 2019;149:706-714
- Laing et al., Trends in Parasitology 2017;33(6):463-472
- Armoo et al., Parasites & Vectors 2017;10:188:1-10
Reviewer 2 Report
Line 54: who survive ivermection treatment etc. Is it a case of the female worms are not affected at all, or ivermectin is not as highly effective against the adults as it is against the mf? If it’s the former perhaps word it to indicate this.
Line 64: Has genetic analysis been done to see if there is a genotype associated with this?
Table 1: any particular reason for the differences between months between pre-treatment snip and post-treatment snip?
Line 169: Was this difference in gender and age to the other areas significant? Looking at the table, I’m guessing not as total numbers are very low. But if you are going to say something is higher or lower, should add in if it’s significant or not. I.e. with the exception of Mahenge where participants appeared to be older and more often female, although this was not significant.
Table 2: Is Mahenge a smaller or less populated region overall? Or just poor recruitment from the area for whatever reason? Or just, overall lower parasite prevalence (mf density seemed very low)? With Maridi and Aketi, the age range is much lower, is there differences in recruitment between the areas? I.e. focus on recruiting high school students more in some areas compared to others?
Table 3: Put PWE definition in the caption as well. Would be useful to have the pre-ivermectin mf density in this table as well for easy comparison
Line 231: This seems like a bit of a random segue. Where is the moxidectin coming from, is this comparing your study with a moxidectin treatment study? Did they look specifically at ppl with epilepsy? I think a bit more of an explainer/introduction to moxidectin is required here – why are you comparing your results with this study? Is this suggesting that moxidectin is a better treatment than invermectin?
Line 268: delete e.g.
Line 268: Do you see the same issues with moxidectin treatment? Would epilepsy drugs affect moxidectin in a similar way?
Reviewer 3 Report
The article is well write. I suggest to improve the introduction and insert othera recent articles such as:
- Targeting Human Onchocerciasis: Recent Advances Beyond Ivermectin (Book Chapter), Sainas, S., Dosio, F., Boschi, D., Lolli, M.L, Annual Reports in Medicinal Chemistry Volume 51, 1, 2018, Pages 1-38
Round 2
Reviewer 1 Report
Most of the issues mentioned in the review of the previous version of this manuscript have been addressed well.
There is one important remaining problem with the description of study participants as described in the text and represented in table 3. In order to avoid misinterpretation of the study’s design and results, further clarification is needed:
To start with Page 2 lines 73-75: “Study participants were persons with epilepsy (PWE) that were identified during door-to-door surveys in four onchocerciasis endemic study sites.” This implies that study participants are ALL women with epilepsy, though within onchocerca endemic study sites, thus WITH or WITHOUT prior mf-positive skin snips. The fact that study sites were endemic for onchocerca does not imply that all women with epilepsy had onchocerca/mf-positive skin snips.
Accordingly, one would expect the denominator for all these participants (PWE), thus n=102, 136, 74, 17 for the respective study sites given in Table 3 to be PWEs with or without prior mf-positive skin snips.
If, however, these denominators in Table 3 are only for women with epilepsy AND prior mf-positive skin snips, then one would like to have - in addition – also the denominator for all women with epilepsy, including those with prior mf-negative skins snips, and split up in with and without.
A flow diagram starting with all study participants, thus including both mf-negative and mf-positive skins snips may help to clarify.
The authors’ cover letter to the revised version of the manuscript (R1) explains that the denominators used include also women who had complete disappearance of mf in skin snips, but this is not specified in Table 3. If so, split out and specify the totals in mf-positives and mf-negatives (those with complete disappearance of mf).
The authors’ cover letter implicitly suggests that the study started only with PWE AND prior mf-positive skin snips, but if indeed the four studies sites had started with door-to-door house surveys one would expect a proportion of PWE to be mf-negative for skin snips, and it would be helpful to understand the numbers and proportions of these initially mf-negative PWEs for each of these study sites.
In summary, a flow diagram specifying the inclusions and exclusions of persons as participants, adding information to table 3, and rewording/adapting/supplementing the text is indispensable.
The title of table 3 may can be more specific too, kin accord with the previous remarks.
